# *PAH* Pathogenic Variants and Clinical Correlations in a Group of Hyperphenylalaninemia Patients from North-Western Romania

**DOI:** 10.3390/diagnostics13081483

**Published:** 2023-04-20

**Authors:** Alin Iuhas, Claudia Jurca, Kinga Kozma, Anca-Lelia Riza, Ioana Streață, Codruța Petcheși, Andra Dan, Cristian Sava, Andreea Balmoș, Cristian Marinău, Larisa Niulaș, Mihai Ioana, Marius Bembea

**Affiliations:** 1Faculty of Medicine and Pharmacy, University of Oradea, 410087 Oradea, Romania; 2Clinical Emergency County Hospital Oradea, 410039 Oradea, Romania; 3Regional Center for Medical Genetics Bihor, Clinical Emergency County Hospital Oradea, 410169 Oradea, Romania; 4Regional Centre of Medical Genetics Dolj, Emergency County Hospital Craiova, 200642 Craiova, Romania; 5Laboratory of Human Genomics, University of Medicine and Pharmacy of Craiova, 200638 Craiova, Romania

**Keywords:** phenylketonuria, *PAH* pathogenic variants, genotype–phenotype correlation, inborn error of metabolism

## Abstract

Phenylketonuria (PKU) is caused by mutations in the phenylalanine hydroxylase (*PAH*) gene and is characterized by altered amino acid metabolism. More than 1500 known *PAH* variants intricately determine a spectrum of metabolic phenotypes. We aim to report on clinical presentation and *PAH* variants identified in 23 hyperphenylalaninemia (HPA)/PKU Romanian patients. Our cohort exhibited classic PKU (73.9%, 17/23), mild PKU (17.4%, 4/23), and mild HPA (8.7%, 2/23). Severe central nervous system sequelae are frequent in our cohort in late-diagnosis symptomatic patients, which highlights yet again the significance of an early dietary treatment, neonatal screening and diagnosis, and facilitated access to treatment. Next-generation sequencing (NGS) identified a total of 11 *PAH* pathogenic variants, all previously reported, mostly missense changes (7/11) in important catalytic domains. c.1222C>T p.Arg408Trp was the most frequent variant, with an allele frequency of 56.5%. Twelve distinct genotypes were identified, the most frequent of which was p.Arg408Trp/p.Arg408Trp (34.8%, 8/23). Compound heterozygous genotypes were common (13/23), three of which had not been previously reported to the best of our knowledge; two correlated with cPKU and one showed an mPKU phenotype. Generally, there are genotype–phenotype correlation overlaps with the public data reported in BIOPKUdb; as our study shows, clinical correlates are subject to variation, in part due to uncontrolled or unknown epigenetic or environmental regulatory factors. We highlight the importance of establishing the genotype on top of using blood phenylalanine levels.

## 1. Introduction

The phenylalanine hydroxylase (PAH) gene encodes a liver enzyme that changes phenylalanine (Phe) to tyrosine in the presence of tetrahydrobiopterin (BH4), molecular oxygen, and iron. The activity of the enzyme is decreased as a result of pathogenic variants, leading to phenylketonuria (PKU; OMIM #261600).

Pathogenic variants in the phenylalanine hydroxylase (*PAH*) gene result in a decrease in the activity of the liver enzyme that converts phenylalanine (Phe) to tyrosine in the presence of tetrahydrobiopterin (BH4), molecular oxygen, and iron. Decreased enzyme activity leads to hyperphenylalaninemia. In rare cases (1–2%), hyperphenylalaninemia may be caused by a deficiency of the cofactor BH4 [1].

PKU prevalence is estimated at 1:24,000 newborns, with significant variations between geographical regions and ethnic groups [2]. The average prevalence in Europe is estimated at 1:10,000 newborns [2]. In Romania, newborn screening for PKU has been implemented since 1979, but only since 2011 has the program had national coverage [3]. Scarce literature works place Romanian incidence estimates in a similar range to the ones mentioned above [4,5,6].

To date, more than 1500 different *PAH* variants are known [7]. The most common types of variants are substitutions, followed by deletions and duplications in about 1/5 of the cases. Missense variants are the most common (58%) [2]. Most patients are compound heterozygous [2]; allelic interaction can complicate the phenotype prediction [8]. In compound heterozygosity, the allele causing a milder phenotype is always dominant to the allele causing severe phenotypes [2].

In most cases, phenotype predictions based on the evaluated genotype are possible [8], although predictive models face several challenges. Splicing, synthesis, or protein are inherited independently of the *PAH* gene. This explains the discordant phenotypes and, ultimately, the variable phenotypes of the same *PAH* variant [9]. The discrepancies noted between the predictive models and previous literature reports are attributed to variants with unpredictable evolution [10], to nosological misclassification of patients resulting from an early collection of samples, or to individual variations in the metabolism of Phe [8].

Depending on the plasma levels of Phe before treatment initiation, PKU is classified as (i) classical PKU (cPKU), (ii) mild PKU (mPKU), and (iii) mild hyperphenylalaninemia (HPA) [8]. Untreated disease develops with irreversible intellectual deficits and a multitude of symptoms such as convulsions, short stature, motor deficits, hypopigmentation, and psychiatric symptoms [1]. Early detection and dietary intervention allow for nearly normal somatic and mental development [11]. Phenylketonuria therapy is largely based on the dietary restriction of phenylalanine.

The aim of the current study is to contribute to the *PAH* mutational spectrum of PKU in patients from North-Western Romania, while assessing correlations between genotypes and metabolic phenotypes, in the context of age of diagnosis, as well as access to and compliance with treatment.

## 2. Materials and Methods

### 2.1. Ethics Statement

Informed consent for this study was obtained from all patients or their legal representative. The Ethics Committee of the Oradea County Emergency Clinical Hospital approval number is 14734/27 April 2021. All of the families received appropriate genetic counseling.

### 2.2. Subjects

The study group consists of 23 patients with HPA/PKU diagnosed between 1981 and 2021 at the Regional Center for Medical Genetics Bihor (CRGM Bihor), Emergency Clinical County Hospital Oradea. The study group consists of 14 female patients (60.9%) and 9 male patients (39.1%), aged between 1 month and 39 years old, with a median of 4 months old. They were unrelated individuals, with the exception of two affected sisters.

The clinical diagnosis of PKU was established based on the plasma hyperphenylalaninemia accompanying suggestive signs and symptoms (in the case of patients diagnosed late) or on the basis of screening tests performed at birth and subsequently confirmed by molecular testing.

This is a sub-study of a previously described cohort [12] including only patients accepting genetic testing.

### 2.3. Molecular Diagnosis

Molecular testing for short variants was performed using the TruSight Inherited Diseases oligo panel with Illumina Rapid Capture library preparation kit for 23 probands. Paired end 2× 150 bp reads on an Illumina NetSeq550 IVD sequencing platform with a median coverage of at least 100× were mapped to the GRCh37 and analyzed using the nf-core/sarek 2.7.1 pipeline. The identified germline variants were annotated using ENSEMBL variant effect predictor (VEP) [13], with several plugins for predictive scores [14]; OMIM, ClinVar, and Varsome were also consulted [15,16,17]. ACMG compliant variant classification was evaluated [18]. A depth threshold of over 20× was considered for diagnostic purposes. Furthermore, the coverage of the *PAH* and *QDPR* genes was manually investigated. We are reporting on pathogenic/likely pathogenic variants identified and classified according to ACMG guidelines.

Variants identified by NGS were confirmed by capillary sequencing. Thirty-six family members agreed to be tested for their carrier status.

### 2.4. Genotype–Phenotype Correlations

The results of the molecular test were compared to data contained in the *PAH*vdb and BIOPKUdb databases (http://www.biopku.org, accessed on 1 March 2023). *PAH*vdb (*PAH* International Database of Variations in Phenylalanine Hydroxylase Gene) [7] is a database containing over 1500 *PAH* variants, as well as data related to allelic phenotype value (APV) or allele frequency. APV is a value that defines the severity of a genotype, which helps establish a genotype–phenotype correlation.

### 2.5. BH4 Responsiveness

The BIOPKUdb database (Database of Patients and Genotypes Causing HPA/PKU) [19] provides data on genotypes, associated metabolic phenotypes, the BH4 response (where reported), or the geographic distribution of these genotypes. We used BIOPKUdb to assess the BH4 responsiveness for the genotypes in our group. Where genotypes or their BH4 responsiveness had not yet been reported, we relied on information available for each allele.

### 2.6. Statistical Analysis

Descriptive statistics (average, minimum, and maximum) were determined for clinical data of patients. Comparisons were performed using MedCalc software version 20.010 (MedCalc Software Ltd., Ostend, Belgium), considering a *p*-value < 0.05 as significant.

## 3. Results

### 3.1. Clinical Description

Table 1 present several clinical characteristics of the studied group.

The age of diagnosis was in most cases in the first months of life; in our cohort, late diagnosis, after the age of 1, was seen in 1/4 of the cases. Delayed diagnosis was to be expected in the pre-screening period; between 1981 and 2011, the average age of diagnosis was 20.64 months; introducing screening policies proved to be efficient as, during 2011–2021, there was a significant drop to an average age of 0.62 months.

According to the metabolic phenotype, the vast majority of the patients have a classic form of the disease—cPKU, with only two patients showing HPA.

Good metabolic control was observed in approximately half of the patients. There was a statistically significant inversely proportional correlation between the patient’s age and good metabolic control (*p* = 0.0002).

Intellectual disability is present in almost half of the cases, with varying degrees of impairment. Unsurprisingly, statistically significant positive correlations were found between the presence of intellectual disability and unsatisfactory metabolic control (*p* < 0.0001) and between the presence of intellectual disability and the age of the patient (*p* = 0.02). In our study, intellectual disability was more likely to be found in older patients. Late diagnosis of the disease correlates with the severity of intellectual impairment (*p* = 0.0001), regardless of the patient’s metabolic phenotype.

### 3.2. PAH Mutation Spectrum

Molecular testing through NGS identified 11 distinct *PAH* variants, reported in Table 2. Missense variants were predominant, followed by frameshift, with relatively few splice variants. The most affected exon was E12, followed by E11.

The most frequently identified variant in the study group was c.1222C>T, p.Arg408Trp. The second-most common *PAH* variant identified was c.1089del, p.Lys363AsnfsTer37. Variants’ frequencies can be found in Table 2.

Most of the identified variants have an APV between 0 and 2 and enzymatic activities between 2% and 44%, being associated with a severe phenotype (cPKU). The only exception is c.898G>T, while p.Ala300Ser *PAH*vdb registers to have an APV of 9.7 and an enzymatic activity of 65%, thus leading to a mild phenotype (HPA).

Two affected sisters were tested and we identified the same haplotype for both cases.

Thirty-six unaffected family members agreed to be tested for their carrier status. In all of the cases, the unaffected parents and/or siblings were carrying one allele that was inherited by the affected offspring.

### 3.3. Genotype–Phenotype Correlation

Most of the patients, 13 (56.5%), were compound heterozygous. Out of the 10 homozygous, there were eight cases of c.1222C>T; p.Arg408Trp; and one case each of c.1089del, p.Lys363AsnfsTer37 and c.842C>T, p.Pro281Leu. None of the 10 homozygous patients had a family history of consanguinity. The remaining genotypes had a lower frequency (see Table 3).

The p. Arg408Trp/p.Arg408Trp genotype was identified in eight patients. Of these, six were cPKU and two were mPKU; five patients had good metabolic control and normal intellect and three patients with poor metabolic control developed intellectual disability.

The p.Arg408Trp/p.Arg158Gln] genotype was identified in three patients. All of these patients have a classical form of the disease (cPKU). Two of them, with good metabolic control, have normal intellect, but one, with poor metabolic control, has a moderate intellectual disability.

Both patients with genotype p.Arg408Trp/p.Leo48Ser have the cPKU metabolic phenotype. One patient had a moderate intellectual disability and one patient had normal intellect.

The p.Lys363AsnfsTer37/p.Arg408Trp genotype was identified in two patients, one with cPKU and a severe intellectual disability and one with mPKU and normal intellect.

Two patients were identified with p.Lys363AsnfsTer37/p.Lys363AsnfsTer37 genotype, both with the metabolic phenotype mPKU.

For the three genotypes we could not identify in the literature, we report the phenotype we identified in our patients.

## 4. Discussion

### 4.1. Screening and Diagnostic Strategies in Romania

Prevalence data are crucial to establish health policies. National registries for PKU are missing in Romania. Our estimates would indicate that the prevalence of PKU in North-Western Romania may be closer to 1:7843 born [12], higher than the national estimate, but comparable to values reported in Eastern European countries [2].

Given the wide age range in our study group, we indirectly capture the benefits of screening efforts in early identification of PKU cases; the age of diagnosis is a valuable indicator and we can see a significant improvement with screening implementation.

Currently, in Romania, screening for phenylketonuria is carried out at the national level in five regional centers, an activity financed by a National Health Program, as in most European countries [20], and follows a methodology established in a guide developed by the Ministry of Health.

Samples are collected 24–72 h after birth, on Guthrie strips [21], in accordance with the recommendations of the European guide [22]. According to the Romanian guide, the dry spot samples will be sent to the regional reference laboratory for analysis within a maximum of 1 week after collection to be analyzed by tandem mass spectrometry (MS/MS). A positive screening test (Phe values above 2 mg/dL associated with normal or low tyrosine values) requires referral of the patient to a diagnostic and treatment center in PKU to confirm the diagnosis by measuring plasma phenylalanine [21].

This procedure, which is partly based on the compliance of the caregivers, makes it improbable to diagnose the disease and initiate the treatment before the age of 1 week, which is the ideal situation [23]. The progression of these events is slower or faster from center to center (Netherlands, 4–7 days; the U.K., 5 days; and Spain, 5–10 days) [24]; the genetic load in each region must also be taken into account; many of the patients from southern Europe with mild phenotypes can remain untreated.

There is unanimity in the literature and among professionals that patients with untreated blood Phe concentrations >600 μmol/L require treatment. In Romania, the value of phenylalanine at which it is recommended to start treatment is 360 mmol/L. European centers introduce treatment at different values of phenylketonuria, but all report values between 360 and 600 μmol/L [25]. There is a consensus at the European level that patients with phenylalanine values below 360 mmol/L should remain untreated [22]. In Romania, children with hyperphenylalaninemia must also be tested for tetrahydrobiopterin deficiency, either by measuring pterins and dihydropteridine reductase activity or by a Sapropterin loading test and, after the age of 4, Sapropterin sensitivity testing is recommended for all patients.

Genetic testing is optional according to the Romanian guide [21]. The European consensus does not consider genotyping essential, but recognizes its importance in determining the degree of protein dysfunction, residual PAH activity, and the predicted metabolic phenotype [22].

### 4.2. Phenotype in Our Study Group

The clinical characteristics of our study group reflect Romania’s healthcare system response to diagnose and treat PKU over the decades. We report a relatively high intellectual disability rate and a rather low percentage of good metabolic control. The severe phenotype (cPKU) was found in most of the patients from our study group, especially in the first two decades studied. The clinical picture of cPKU patients consists of early, severe, and progressive neurological manifestations, with symptomatology worsening with the delay of the dietary treatment.

The milder metabolic phenotype was found in a small percentage of patients, notably lower than the averages reported in the literature (21.9%) [2]. In the absence of screening tests or significant clinical manifestations, it can be assumed that patients with mild forms of the disease remained undiagnosed, which would contribute to the overall low percentage of patients with *PAH* in the study group.

National screening limitations are responsible for the varying diagnostic rates over time and, consequently, with varying interventions. This explains the different background we see in our patients; that is, a mild phenotype in rigorously managed treatment cases, who are socially integrated, as well as a severe phenotype in low compliance cases, from disadvantaged backgrounds.

Despite dietary restriction being undeniably successful, there are still unsolved issues; that is, adherence to treatment progressively decreases with age, reflected in the poor control of the disease in adolescence [26,27]. Good dietary control in childhood followed by reduced compliance later may lead to behavioral and social challenges [28].

The onset of intellectual disabilities in patients over 20 years old is mainly due to the lack of medical food in sufficient quantities at pediatric age. It was also observed in our study that one patient with cPKU, despite the good metabolic control, shows a mild intellectual impairment.

In one of our cases, despite rapid diagnosis and good control of serum Phe levels, we have seen a persistence of neurological problems and decreased quality of life.

Altogether, our findings emphasize, in line with the boon of already published literature, the importance of early therapeutic intervention, preferably in the first week [13], to favor normal neurodevelopment [14].

### 4.3. Genotype in Our Study Group

In counselling, detecting the *PAH* genotype is important for inferring the metabolic progression, especially in the context of the genetic makeup of each ethnicity. More than two-thirds of the genotypes correlated to the cPKU phenotype in the present study.

The NGS-based approach allows us to identify *PAH* genotypes with short variants in patients initially classified as HPA/PKU; however, in cases with persistent HPA, the identification of microdeletions or microduplications, using, for instance, multiplex ligation- dependent probe amplification (MLPA) analysis, could be an option. To date, there are no available data to estimate the frequency of large rearrangements, which sequencing does not cover well or at all [2].

Causative variants reported in our study cluster in exons that encode the catalytic *PAH* region, especially exons 12 and 11, and have all been previously reported.

Notably, we identified the “Celtic” c. 1222C>T, p.Arg408Trp pathogenic variant in most of the patients. This variant is most frequently reported in all populations, especially in Central and Eastern Europe (44.4–53.7%) [2]. According to Hillert et al. [2], the variant p.Arg408Trp is found with decreasing frequencies from east to west, indicating a pattern of population migration.

Three genotypes had not previously been reported in the BIOPKUdb database: p.Lys363AsnfsTer37/p.Lys363AsnfsTer37, p.Arg413Pro/p.Arg408Trp, and p.(=)/IVS12+1G>A/p.Lys363AsnfsTer37. The genotypes were identified in one patient each. The patient with the p.Lys363AsnfsTer37.p.Lys363AsnfsTer37 genotype showed the mPKU phenotype; the other two showed the cPKU phenotype. We believe this to be relevant in the context of the uniqueness of each ethnicity.

In terms of genotype–phenotype correlations, most of the patients’ phenotypes overlapped with public data reports. Several of the clinical characteristics are described above. The p.[Leu48Ser];[(Gln355_Tyr356insGlyLeuGln)/IVS10-11G>A] genotype was identified in a patient with a Phe value of 8.2 mg/dL at diagnosis. This is in discordance with the data reported in BIOPKUdb, with this genotype generating cPKU (44.19%) or mPKU (55.81%) phenotypes. One can speculate that some of the other discrepancies may be caused by late diagnosis, and any interpretation should be reserved.

In our study, the genotype/phenotype concordance and overlap with BIOPKUdb data was high. BIOPKUdb is a very useful tool to query reported information on *PAH* mutational spectrum and clinical correlates. Despite the limited number of participants included in our study, we can see unique genotypes, especially including rarer variants. Interestingly, in our study, discordance was mainly seen in situations where the predicted phenotype underestimated the clinically evaluated disease severity. This is a remaining limitation of using only blood Phe levels for predictive modelling. We strongly feel that it is important to thoroughly evaluate the genetic makeup in all HPA/PKU patients and report it alongside other data [29].

## 5. Conclusions

PKU is one of the inborn errors of metabolism with high incidence, already introduced in many national screening programs. Subsequent patient management, successful outcome, and even family planning all benefit from early and accurate genetic diagnosis.

The urgency of early dietary intervention is once again emphasized by the high number of serious CNS complications in late-diagnosis patients in our group.

We report c. 1222C>T, p.Arg408Trp as the most frequent variant found in our Romanian study group. It was associated mostly with cPKU. Compound heterozygous genotypes were common, with p.Arg408Trp/p.Arg408Trp being the most common haplotype.

## Figures and Tables

**Table 1 diagnostics-13-01483-t001:** Phenotypic features of our cohort.

Clinical Phenotype	
gender distribution	14 F/9 M
age at diagnosis (median months)	2 (F)/7 (M)
metabolic phenotype	73.9%, 17/23 cPKU
	17.4%, 4/23 mPKU
	8.7%, 2/23 HPA
intellectual disability	43.5%, 10/23
seizures	21.7%, 5/23
behavioral disturbance	52.2%, 12/23
other health issues	30.4%, 7/23
familial history of PKU	13.0%, 3/23
parental consanguinity	no
genetic results	10 homozygous genotypes/ 12 compound heterozygous genotypes

F—females, M—males.

**Table 2 diagnostics-13-01483-t002:** Mutation spectrum of the studied group.

*PAH* Variant NM_000277.3	Location, Variant Type	ACMG Pathogenicity *	Allele Frequency (n = 24)	APV **	Enzymatic Activity **
c.1222C>T p.Arg408Trp	Substitution Missense E12/catalytic	Pathogenic (PP5, PS3, PM1, PM5, PM2, PP3)	56.5%, 26/46	0	19.2%
c.1089del p.(Lys363AsnfsTer37)	Deletion Frame shift E11/catalytic	Pathogenic (PVS1, PP5, PM2)	10.9%, 5/46	0	unknown
c.842C>T p.Pro281Leu	Substitution Missense E7/catalytic	Pathogenic (PS3, PP3, PM1, PM5, PP5, PM2)	6.5%, 3/46	0	2%
c.473G>A p.Arg158Gln	Substitution Missense E5/catalytic	Pathogenic (PP5, PS3, PM1, PP3, PM5, PM2)	6.5%, 3/46	1	10%
c.143T>C p.Leu48Ser	Substitution Missense E2/regulatory	Pathogenic (PP5, PS3, PM1, PP3, PM2)	6.5%, 3/46	2	39%
c.1315 + 1G>A p.(=)/IVS12 + 1G>A	Substitution Splice Intron 12	Pathogenic (PVS1, PP5, PM2)	2.2%, 1/46	0	unknown
c.1066-11G>A p.(Gln355_Tyr356insGlyLeuGln)/IVS10-11G>A	Substitution Intron 10	Pathogenic (PS3, PP5, PP3, PM2)	2.2%, 1/46	0	5%
c.1238G>C p.Arg413Pro	Substitution Missense E12/oligomerization	Pathogenic (PP5, PS3, PM1, PM5, PP3, PM2)	2.2%, 1/46	0	11%
c.782G>A p.Arg261Gln	Substitution Missense E7/catalytic	Pathogenic (PP5, PS3, PM1, PM5, PM2)	2.2%, 1/46	1.6	44%
c.898G>T p.Ala300Ser	Substitution Missense E8/catalytic	Pathogenic (PS3, PP5, PS3, PM1, PM5, PM2)	2.2%, 1/46	9.7	65%
c.168 + 5G>C	Substitution Splice Intron 2	Pathogenic (PP5, PP3, PM2)	2.2%, 1/46	0	unknown

* ACMG classification, ClinGen PAH Expert Panel consulted; ** according to *PAH*vdb (http://www.biopku.org accessed on 1 March 2023).

**Table 3 diagnostics-13-01483-t003:** Metabolic phenotype for our North-Western Romania patient group.

Genotype	Frequency	Predicted Metabolic Phenotype *	Metabolic Phenotype
p.Arg408Trp/p.Arg408Trp	34.8%, 8/23	**cPKU 99.31%** mPKU 0.69% HPA 0.00%	3/8 cPKU 2/8 mPKU
p.Arg408Trp/p.Arg158Gln	17.4%, 3/23	**cPKU 100%** mPKU 0.00% HPA 0.00%	3/3 cPKU
p.Arg408Trp/p.Leu48Ser	8.7%, 2/23	**cPKU 68.33%** mPKU 31.67% HPA 0.00%	2/2 cPKU
p.Lys363AsnfsTer37/p.Arg408Trp	8.7%, 2/23	**cPKU 100% **** mPKU 0.00% HPA 0.00%	½ cPKU ½ mPKU
p.Lys363AsnfsTer37/p.Lys363AsnfsTer37	4.4%, 1/23	no records of this genotype	1/1 mPKU
p.Arg408Trp/p.Arg261Gln	4.4%, 1/23	**cPKU 76.04%** mPKU 23.61% HPA 0.35%	1/1 cPKU
p.(=)/IVS12+1G>A/p.(Lys363AsnfsTer37	4.4%, 1/23	no records of this genotype	1/1 cPKU
p.Arg408Trp/p.(=)/IVS2+5G>C	4.4%, 1/23	**cPKU 100%** mPKU 0.00% HPA 0.00%	1/1 cPKU
p.Pro281Leu/p.Pro281Leu	4.2%, 1/23	**cPKU 99.21%** mPKU 0.79% HPA 0.00%	1/1 cPKU
p.Leu48Ser/p.Gln355_Tyr356insGlyLeuGln)/IVS10-11G>A	4.4%, 1/23	cPKU 44.19% **mPKU 55.81%** HPA 0.00%	1/1 HPA
p.Arg413Pro/p.Arg408Trp	4.4%, 1/23	no records of this genotype	1/1 cPKU
p.Pro281Leu/p.Ala300Ser	4.4%, 1/23	cPKU 0.00% mPKU 6.25% **HPA 93.75%**	1/1 HPA

* according to BIOPKUdb (http://www.biopku.org accessed on 1 March 2023); ** only one previous reported case.

## Data Availability

Not applicable.

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
