# Peer review of "PAH Pathogenic Variants and Clinical Correlations in a Group of Hyperphenylalaninemia Patients from North-Western Romania"

_diagnostics, 2023, doi:10.3390/diagnostics13081483_

Round 1
Reviewer 1 Report
Review the nomenclature of the pathogenic variants as it differs for the same variant in different parts of the text
The clinical description of the patients requires more information as, for example, it is not clear if there are any familial cases. A figure with pedigrees and associated variants would be helpful to understant the results.
The authors do not describe if they used other methods to confirm the findings, also they state that genetic assessment was offered but it is not described if they carried out segregation analysis in the families.
There is no further functional analysis of the three new pathogenic variants identified, for example in silico analysis of the pathogenic variant to explain the effect in the protein and how it affects its function and its correlation with the phenotype.
Author Response
Dear reviewer,
We are grateful for the time and effort that you and the other reviewers dedicated to providing feedback on our manuscript and are thankful for the insightful comments on and valuable improvements to our paper.
We have made all efforts to address all the concerns you raised. We have incorporated most of the suggestions made. The changes can be found in the track-changes new manuscript. We appreciate all your suggestions, and tried our best to explain our thoughts. Please see below for a point-by-point response to your comments and concerns, marked in blue.
Best wishes,
The authors.

Reviewer 2 Report
This paper reports on the clinical presentation and PAH variants identified in 23 hyperphenylalaninemia (HPA)/PKU Romanian patients, with a focus on genotype-phenotype correlations. They found svere CNS sequelae in late-diagnosis cases. NGS identified 11 pathogenic PAH variants, with c.1222C>T, p.Arg408Trp being the most common. Compound-heterozygous genotypes were common. The study emphasizes the importance of early diagnosis and genotype determination to understand clinical outcomes.
Overall, the manuscript presents a well-defined research objective, robust methodology, and sound results, with a praiseworthy review of the pertinent literature. However, the study's validity is subject to limitations given the small sample size, potentially constraining the generalizability of the conclusions. Furthermore, there is a lack of clarity regarding the methods and outcomes of HPA/PKU genotyping and phenotyping in Romania, and its contributions to the scientific community are somewhat obscure. To address this issue, the authors may consider supplementing the article with a comparative table or a dedicated section discussing the similarities and differences with other countries globally, thereby bolstering its scientific impact and appeal.
Author Response

(The authors gave the same response as above.)
